# Combined C-Reactive Protein and Novel Inflammatory Parameters as a Predictor in Cancer—What Can We Learn from the Hematological Experience?

**DOI:** 10.3390/cancers12071966

**Published:** 2020-07-19

**Authors:** Øystein Bruserud, Helene Hersvik Aarstad, Tor Henrik Anderson Tvedt

**Affiliations:** 1Section for Hematology, Institute of Clinical Science, Faculty of Medicine, University of Bergen, 5007 Bergen, Norway; Helene.Aarstad@uib.no; 2Section for Hematology, Department of Medicine, Haukeland University Hospital, 5021 Bergen, Norway; tor.henrik.anderson.tvedt@helse-bergen.no

**Keywords:** inflammation, cancer, prognosis, acute phase reaction, C-reactive protein, leukocyte counts, monocyte, cytokine, neutropenia, allogeneic stem cell transplantation

## Abstract

The acute phase reaction is a systemic response to acute or chronic inflammation. The serum level of C-reactive protein (CRP) is the only acute phase biomarker widely used in routine clinical practice, including its uses for prognostics and therapy monitoring in cancer patients. Although Interleukin 6 (IL6) is a main trigger of the acute phase reactions, a series of acute phase reactants can contribute (e.g., other members in IL6 family or IL1 subfamily, and tumor necrosis factor α). However, the experience from patients receiving intensive chemotherapy for hematological malignancies has shown that, besides CRP, other biomarkers (e.g., cytokines, soluble cytokine receptors, soluble adhesion molecules) also have altered systemic levels as a part of the acute phase reaction in these immunocompromised patients. Furthermore, CRP and white blood cell counts can serve as a dual prognostic predictor in solid tumors and hematological malignancies. Recent studies also suggest that biomarker profiles as well as alternative inflammatory mediators should be further developed to optimize the predictive utility in cancer patients. Finally, the experience from allogeneic stem cell transplantation suggests that selected acute phase reactants together with specific markers of organ damages are useful for predicting or diagnosing graft versus host disease. Acute phase proteins may also be useful to identify patients (at risk of) developing severe immune-mediated toxicity after anticancer immunotherapy. To conclude, future studies of acute phase predictors in human malignancies should not only investigate the conventional inflammatory mediators (e.g., CRP, white blood cell counts) but also combinations of novel inflammatory parameters with specific markers of organ damages.

## 1. Introduction

Serum levels of C-reactive protein (CRP) have been used for several decades as a marker of the acute phase reaction that can be induced both by acute and chronic states as a systemic reaction to inflammation, injury, or infections [1,2]. However, the acute phase reaction alters circulating levels of a myriad of inflammatory mediators [3,4,5], including certain cytokines that can function as inducers or drivers of the reaction and/or can be released during the later phase [1]. CRP is the only acute phase biomarker that is used in routine clinical practice [3,6,7], whilst several circulating biomarkers of inflammation have been characterized during the last decade. These new parameters may serve as an alternative to CRP or should be combined. In the present article, we review the hematological experience with new acute phase biomarkers and how these new markers have been used in hematological malignancies.

## 2. The Acute Phase Reaction and the Biology of CRP

### 2.1. The Acute Phase Reaction

The acute phase reaction is characterized by increased levels (i.e., at least 25% increase in serum levels) of several proteins in response to inflammation, infection, or tissue injury, but the name is misleading because the reaction can also be seen in chronic diseases with fluctuating conditions (e.g., cancers), or be a chronic or long-lasting response that is maintained during the chronic disease [2,3,6]. The reaction is regarded as a response induced by cytokines produced at inflammatory sites; Interleukin 6 (IL6) has been referred to as the chief stimulator but several other mediators can also contribute to the induction of the response including other members of the IL6 cytokine family, IL1β, TNFα, IFNγ, TGFβ, and IL8/CXCL8 [1]. Furthermore, studies in knockout mice suggest that the relative contribution/importance of various inducer cytokines depends on the nature and/or the site of inflammatory [1]. Finally, the effects of various cytokine combinations may have additive, synergistic, or inhibitory effects on the levels of individual acute phase proteins [1]. Taken together, these experimental observations suggest that the overall acute phase protein profile depends on the nature of the initial inflammatory event, how this local event then induces a systemic (i.e., serum/plasma) protein/cytokine response and finally how variations in the response-inducing signaling modulate the release and thereby the overall profile of the various acute phase proteins.

The acute phase reaction was initially described as a reaction involving various serum proteins, including increased levels of several complement factors, coagulation factors, antiproteases, transport proteins, CRP, serum amyloid A, ferritin, as well as proinflammatory cytokines [1,3]. However, it is also well-known that decreased levels of several proteins, e.g., albumin, transferrin, and IGF1, can be a part of the acute phase reaction and be associated with altered peripheral blood cell counts (i.e., total leukocyte or leukocyte subset, erythrocytes, platelets) [1,4,5].

According to the definition of acute phase proteins many cytokines, soluble cytokine receptors, and soluble adhesion molecules can be classified as acute phase proteins. The final acute phase reaction (i.e., the overall effect on serum protein levels) reflects both the initial step/mechanisms for induction of the reaction (see above) and altered serum levels of associated proteins responding to the inducers. However, the induction and the reaction proteins are overlapping, as can be illustrated by the CRP example. Serum CRP production and release can be induced by IL6, IL8, and TNFα [7], but at the CRP itself can thereafter increase the release of these proinflammatory mediators both at the local sites of inflammation [8] and thereby also their systemic levels [9] (see Section 2.2).

The complexity of the acute phase reaction is illustrated by the observations from patients with febrile chemotherapy-induced neutropenia (Table 1). Even these severely immunocompromised patients show an acute phase reaction with increased serum CRP levels in response to infections [10], and they also show significantly altered levels of a wide range of other mediators, especially regulators of inflammation as well as several other biomarkers [10,11,12,13,14,15,16,17,18,19,20,21,22,23,24]. CRP is the only marker that is widely used in the routine clinical evaluation of these patients. However, the overall results reviewed above suggest that combined use of several mediators or more complex acute phase protein/cytokine profiles is an alternative to the use of CRP alone to detect the acute phase reaction.

A similar complexity of the acute phase reaction is also observed for immunocompetent individuals. This is illustrated by a recent study investigating patients with sepsis [25]. The authors compared serum levels of 35 soluble mediators (i.e., 16 cytokines, six growth factors, four adhesion molecules, nine proteases/protease inhibitors) for sepsis patients with and without bacteremia. Although sixteen mediators differed significantly between the groups, the two patient subsets could be identified by a biomarker profile including six mediators. Thus, although the acute phase reaction reflects complex inflammatory mechanisms, a profile only including six carefully selected mediators could be used for clinical evaluation of the patients.

### 2.2. The Biology of CRP

The molecular structure and the biological functions of CRP have been described in several recent reviews [7,9,26,27,28,29,30], and for this reason we only include a relatively brief description in the present article. The most important molecular and biological characteristics of CRP are summarized in Table 2.

CRP exists in several isoforms [9,26,27]. The basic molecular unit of all isoforms is the monomeric CRP molecule (mCRP) with 206 amino acids and a molecular weight of 23 kDa; this is normally a nonglycosylated protein. The monomer includes one disulfide bond and has a ligand-binding surface that can bind two calcium ions. The plasma CRP is mainly the pentameric isoform of hepatic origin. The effects of pentameric CRP on monocyte/macrophages/neutrophils are mainly mediated by the complement receptors CD32, CD64, and CD89, but effects on endothelial cells and platelets can be mediated by the low-density lipoprotein receptor and α_v_β_3_ integrins, respectively. Monomeric CRP effects can also be mediated via lipid rafts, and integrins than α_v_β_3_ can also bind monomeric CRP. Furthermore, recent studies suggest that CRP can also exist as fibril-like structures [31] as well as trimers, tetramers, and decameres [32,33]. Finally, CRP may become less resistant to proteolysis after structural rearrangements, and peptides with biological activity may then be formed [28].

Infusion of recombinant human CRP into healthy volunteers results in significantly increased serum levels of the proinflammatory cytokines IL6 and IL8 as well as amyloid A, phospholipase A, and several coagulation parameters including prothrombin 1 and 2 [34]. These observations show that CRP is not only a marker of inflammation, but a proinflammatory mediator that participates in the development of inflammation; pharmacological targeting of CRP has, therefore, been suggested as a possible therapeutic strategy in inflammatory diseases [35].

CRP should be regarded as an immunoregulatory or immunomodulatory mediator. As can be seen from Table 2 it can alter the function of a wide range of cells, including several immunocompetent cells, indirectly, through altered levels of a wide range of immunoregulatory mediators. It can also directly influence the function of immunocompetent cells, especially cells in the innate immune system.

### 2.3. Summarizing Comment: The Acute Phase Reaction Reflects Complex Biological Mechanisms and the Acute Phase Protein Profile Will Probably Differ Both between Individual Patients and between Inflammatory Diseases

Taken together the results described above illustrate that the acute phase reaction reflects complex inflammatory events. The acute phase response can be detected both for immunocompetent and severely immunocompromised cancer patients with severe leukopenia due to bone marrow failure caused by the intensive chemotherapy. These events induce altered levels (increased and/or decreased) of a wide range of biologically diverse mediators. The overall mediator profile of the acute phase reaction probably depends on the nature and localization of the inflammatory process as well as individual patient characteristics (e.g., the influence of polymorphisms in the CRP gene). Secondary effects on the normal peripheral blood cell counts can also be present, possibly due to the altered levels of several hematopoietic growth factors and other regulators of hematopoiesis during the acute phase reaction. A more detailed and careful analysis of the acute phase reaction would therefore be to investigate combinations of mediators (e.g., selected cytokines, cytokine families, inducers of the acute phase reaction), and such profiling may even be helpful in the clinical evaluation of patients with identification of patient subsets based on biological characteristics. In this context, CRP should be regarded as a marker of inflammation, but it will probably reflect only a part of the molecular mechanisms involved in the development of an acute phase reaction in individual patients.

## 3. Proinflammatory Markers for Prognostication in Cancer Patients; Molecular and Hematological Markers

### 3.1. Inflammaging; Inflammation as a Part of the Aging Process

Inflammation is regarded as a hallmark of aging and this is often referred to as inflammaging, i.e., a smoldering proinflammatory phenotype that accompanies aging [36]. It is characterized by increased production of IL1β as well as TNFα, two mediators that are involved in the induction of the acute phase reaction [1]. Inflammaging is probably a multifactorial process that may be caused by proinflammatory tissue damage, immunological dysfunction, increased cytokine secretion by senescent cells, increased activation of proinflammatory transcription factors (e.g., NFκB), and/or defective autophagy [37].

A recent study investigated inflammatory responses in healthy stem cell donors [38]. This study included a consecutive group of 98 healthy individuals (median age 49 years, range 18–77 years) that were accepted as allogeneic stem cell donors; i.e., they were regarded as healthy after a careful medical evaluation. Their CRP levels were generally low and 75% of them had CRP level < 2 mg/L; 50% of them had CRP levels below the detection level of 1 mg/L. Increased CRP levels were detected especially in the elderly donors, and the increased levels seemed to be maintained over time. These increased CRP levels may thus reflect inflammaging. Furthermore, population studies have shown an association between CRP level and both all-cause mortality, cancer mortality and cardiovascular mortality [39,40], and CRP levels also show associations with frailty (i.e., a geriatric syndrome characterized by a physiological state of vulnerability) [41]. Furthermore, proinflammation is also associated with obesity and type 2 diabetes [36,41].

Cancer is mainly a disease of the elderly; for the elderly cancer patients, increased CRP levels may not only reflect their malignant disease, but may also be a part of the (normal) aging process. The interpretation of increased serum CRP levels in elderly cancer patients should therefore be careful because aging, frailty, and other disorders may all contribute to the increased CRP levels.

### 3.2. Inflammatory Markers in Solid Tumors

Various biomarkers have been used as markers of proinflammatory activity in cancer patients, the most commonly used are listed in Table 3 [42,43,44,45]. Increased CRP levels or decreased albumin levels are both characteristics of the acute phase response (see Section 2.1) and both have been used as single markers and as the CRP: albumin ratio, also referred to as the Glasgow Prognostic score. This scoring system has been investigated in two different versions, and for both of these versions, CRP above 10 mg/L and albumin below 35 g/L is scored as 1. The original The Glasgow Prognostic score is based simply on the independent scoring of CRP and albumin, and a patient could then get a score ranging from 0 to 2. The Modified Glasgow Prognostic Score is also based on the same cutoffs for CRP and albumin, but CRP below and albumin above the threshold gives a score of 0, increased CRP alone gives a score of 1, and increased CRP together with decreased albumin gives a score of 2.

An alternative to CRP/albumin is to use peripheral blood counts of normal cells (i.e., total leukocytes or leukocyte subsets, platelets) either as the absolute levels (i.e., cell concentrations) of individual cell populations or as a cell count ratio (e.g., lymphocyte:neutrophil ratio). One possibility is to use the count of a myeloid cell population as the prognostic marker, e.g., total white blood cell (WBC), lymphocyte, neutrophil, monocyte or platelet count. The alternative is to use a ratio, i.e., one of the three myeloid cell populations relative to the lymphocyte count. All these alternatives have been used in various studies [43].

The results from systematic reviews and meta-analyses when using all these molecular and cellular parameters as prognostic biomarkers in human malignancies are summarized in Table 3. In these large meta-analyses, they included all suitable studies (i.e., studies fulfilling defined criteria) where the given biomarker was used for prognostication of cancer patients; this means that they included patients with different malignancies and studies from different countries. Most of the included studies were retrospective, and it can be seen from the table that all these markers had (often independent) prognostic impact across type of malignancy and geographical location. It can be seen that the most commonly investigated pro-inflammatory prognostic markers were CRP, albumin, the CRP:albumin ratio, and the neutrophil:lymphocyte ratio.

One problem with the analyses presented in Table 3 is the variation between studies in the thresholds reported to have a prognostic impact. This was a problem especially for CRP and albumin, whereas there was less variation for the thresholds reported for the neutrophil:lymphocyte ratio and especially for the CRP:albumin ratio [43].

Reactive agranulocytosis, monocytosis, or thrombocytosis is common and can be caused by many different conditions, including malignant diseases but also several non-malignant conditions including injury or chronic inflammatory or autoimmune disorders. Many studies have described associations between peripheral blood cell counts and CRP/albumin levels; this is not surprising because acute phase cytokines are important drivers of the acute phase reactions and several hematopoietic growth factors are parts of the acute phase reaction and therefore show increased levels in response to inflammation/infection (see Table 1).

In Table 4, we have summarized the results from systematic reviews and meta-analyses of proinflammatory markers used for prognostication of various human malignancies. As would be expected from the results presented in Table 3 most of these systematic reviews were also based on or included CRP as the proinflammatory prognostic biomarker. It can be seen that CRP levels are associated with prognosis for a wide range of solid tumors. Furthermore, the prognostic impact of proinflammatory biomarkers has been documented in several studies especially for urological cancers but also for breast and colorectal cancer.

To conclude, the results summarized in Table 3 and Table 4 clearly illustrate that proinflammatory markers can be used for prognostic evaluation of cancer patients, and this is true for many different malignancies. The overall results also show that elevated proinflammatory markers are generally associated with adverse prognosis and decreased survival. However, major remaining problems are the wide variation in the threshold between various studies for many of these markers, and it is also difficult to judge which marker should be preferred for the various malignancies. A combination of various markers may be a better solution than the use of single markers, but the problems of threshold and preferred marker still remain. Thus, the prognostic impact of proinflammatory markers in human malignancies is well documented, but the next step should probably be to investigate the molecular and biological mechanisms behind the proinflammatory responses, and to identify the optimal biomarkers for prognostication in various malignancies based on the molecular mechanisms behind the inflammation, and not based on the parameters being easily available from routine clinical practice. It should also be emphasized that most of the markers described in Table 1 can be analyzed by established and easily available methodological strategies.

### 3.3. Inflammatory Markers in Multiple Myeloma

The inflammatory markers can be divided into two main subsets; (i) molecular markers measured in serum or plasma (e.g., CRP, albumin), and (ii) peripheral blood normal cell counts (e.g., various myeloid as well as lymphoid cells) (Table 1, Table 2, Table 3 and Table 4). The peripheral blood normal cell counts are more difficult to use for prognostication in patients with hematological malignancies because these diseases will often have bone marrow involvement that will influence the interpretation of these cell counts. These patients may have bone marrow failure due to the marrow infiltration of malignant cells with decreased peripheral blood normal cell concentrations, but patients with chronic myeloproliferative neoplasias will usually have increased levels of mature myeloid cells (i.e., erythrocytes/leukocytes/platelets) as typical signs of their malignant disease. However, multiple myeloma is an example showing that the commonly used proinflammatory markers can even be used in hematological malignancies with bone marrow involvement although the interpretation of the peripheral blood cell counts (i.e., the platelet count) has to be modified. First, a recent study showed that a combination of high neutrophil:lymphocyte ratio, high CRP level and low platelet counts (i.e., indicating severe bone marrow involvement/failure) at the time of first diagnosis was associated with adverse prognosis and shortened overall survival. This impact was independent of age, renal function, and the International Staging System [66]. The prognostic impact of the neutrophil:lymphocyte ratio was also demonstrated in another myeloma study [67]. Secondly, a large study including 224 patients that were diagnosed during the last decade and received autologous stem cell transplantation (median age 59 years) showed that the neutrophil: lymphocyte ratio (i.e., the Glasgow prognostic score) was the only independent predictor of both overall and progression-free survival [68]. The platelet:lymphocyte score and B symptoms were also independent predictors of progression-free survival whereas high-risk cytogenetics was an independent predictor of overall survival. Third, the Myeloma Research Alliance Risk Profile has also been developed for prognostication of transplant-ineligible myeloma patients; this risk score also included CRP level together with WHO performance status and the International Staging System [69]. The preoperative CRP level was also predictive for the postoperative survival after surgery for myeloma bone disease [70]. Finally, CRP may also be used as a biomarker for response to new, targeted therapy, in multiple myeloma [71].

To conclude, proinflammatory biomarkers can also be used for prognostic evaluation of hematological malignancies with bone marrow involvement, but the interpretation of normal peripheral blood cell counts may need modification due to the bone marrow involvement.

### 3.4. Proinflammatory Markers as Prognostic Biomarkers in Renal Cancer and Squamous Cell Head and Neck Cancer; the Use of Acute Phase Cytokine Profiles Rather Than Single Molecular Markers

We have recently characterized preoperative CRP together with acute phase cytokine biomarkers in patients with renal cancer [72]. Initial studies suggested that serum IL33Rα (soluble Interleukin 33 receptor α chain) levels were associated with prognosis, although its impact was dependent on the overall IL6 cytokine family profile. We therefore examined an extended cytokine profile included seven IL6 family members (IL6, IL6 receptor α, gp130, IL27, IL31, ciliary neurotropic factor/CNTF, oncostatin M/OSM), together with two IL1 subfamily members (IL1RA, IL33Rα) and TNFα. Based on unsupervised hierarchical clustering analysis, we could identify a patient subset with adverse prognosis and a serum cytokine profile especially characterized by high levels of IL6, IL33Rα, and TNFα. Thus, the acute phase cytokine profile and its prognostic impact differ between renal cancer patients. Furthermore, we also investigated the same preoperative 10-cytokine profile in patients with head and neck squamous cell carcinoma [73]. The cytokine profile differed considerably also between these patients, and increased CRP and IL6 levels were independent markers for adverse prognosis (i.e., cancer-related death). The same was true for IL1RA for human papilloma virus (HPV) negative patients and for CNTF for HPV positive patients. Thus, the acute phase reaction also differed between head and neck cancer patients but for these patients the single cytokines seemed to be more important for the prognosis than the 10-cytokine profile.

## 4. The Use of Proinflammatory Markers in Cancer Patients with Immune-Mediated Complications; Studies of Graft Versus Host Disease and Cancer Immunotherapy

### 4.1. Graft Versus Host Disease in Patients with Hematological Malignancies Treated with Allogeneic Stem Cell Transplantation

Allogeneic stem cell transplantation is used in the treatment of young and fit patients with the most aggressive hematological malignancies, and graft versus host disease (GVHD) is one of the most common and severe posttransplant complications and also an important cause of non-relapse mortality [74,75]. Patients at risk of acute GVHD can be identified based on pretransplant prognostic parameters, e.g., age, donor-recipient mismatch, extensive pretransplant anticancer treatment, but a recent meta-analysis also identified increased pretransplant CRP levels as a risk factor for severe post-transplant immune-mediated complications [76]. The molecular mechanisms behind this prognostic impact are not known, but a recent study suggested that increased IL6 systemic levels is a part of this proinflammatory phenotype, and the IL6 family member IL31 may also contribute together with HGF, granulocyte colony stimulating factor (G-CSF), bFGF, and sTNFRI [77,78]. These proinflammatory effects may also influence endothelial cells; this effect is reflected in the association between pretransplant IL6 systemic levels, early posttransplant fluid retention, and increased risk of acute GVHD, transplant-related mortality and overall survival [77]. Previous studies have also described a significant association between pre-engraftment CRP increase not caused by documented infection and increased risk of later GVHD as well as non-relapse mortality [79]. There is also a significant association between the CRP levels in the first three days posttransplant and later engraftment syndrome as well as severe GVHD [80]. Finally, the magnitude of the decline in serum albumin level from the start of conditioning until its nadir level during the first two weeks posttransplant is also associated with later development of severe GVHD [81]. Taken together all these observations suggest that pretransplant and early posttransplant inflammatory events are important for the risk of later immune-mediated complications. However, various acute phase mediators may reflect different characteristics of the acute phase response in allotransplant recipients because the associations between pretransplant CRP and ferritin levels and late outcome after transplantation (i.e., transplant-related mortality and relapse rate) reached statistical significance in multivariate analyses only for pretransplant ferritin but not CRP [82].

Recent studies have demonstrated that healthy stem cell donors are heterogeneous with regard to CRP levels, and especially elderly donors, show increased CRP levels; these levels are further increased during stem cell mobilization by G-CSF. Healthy donors are also heterogeneous with regard to G-CSF mobilization of various immunocompetent cell subsets, and the graft content of immunocompetent cells and the cytokine levels in the graft supernatants thereby show wide variations between healthy stem cell donors [83,84]. However, further studies are needed to clarify whether these differences in donor inflammatory activity and/or graft content of immunocompetent cells or cytokines are important for recipient outcome after allotransplantation [85].

Recent studies have also evaluated the use of CRP for prognostication in allotransplant recipients who develop acute GVHD. One study described higher CRP levels in steroid-refractory patients when they were tested after one week of steroid treatment, and this translated into later increased transplant-related mortality [86]. Proinflammatory markers based on normal peripheral blood cell counts did not have any significant associations with prognosis in this study. Another study also investigated proinflammatory markers after seven days of steroid therapy, and low serum albumin levels together with increased CRP levels were then associated with increased transplant-related mortality also in this study [87]. Thus, CRP seems to be associated with an adverse prognosis in acute GVHD.

Several studies have investigated the possible use of systemic (i.e., serum or plasma) levels of soluble mediators for the diagnosis of acute GVHD. One of these approaches was to investigate the levels of 120 soluble mediators, including several of the mediators listed in Table 1, at the onset of clinical symptoms consistent with acute GVHD [88]. They concluded that four mediators optimally discriminated between patients with and without GVHD, i.e., IL2Rα, TNFRI, CXCL8/IL8, and HGF. Their observations also suggested that this biomarker profile could predict survival independent of GVHD severity. Acute GVHD is usually seen in skin, liver and gastrointestinal tract, and in a later study these authors therefore combined their four biomarkers of inflammation with the skin-specific marker elafin and the gastrointestinal marker regenerating islet-derived 3-α (Reg3α) [89]. This study showed that this six-biomarker panel examined at the time of diagnosis as well as two and four weeks into the treatment predicted nonresponse after four weeks as well as mortality on day +180 posttransplant. Thus, a simple profile including only a limited number of inflammatory biomarkers and possibly combined with organ-specific markers can be used to diagnose and also predict the prognosis of a complex and often multi-organ immune-mediated complication.

An even simpler and more recent strategy is the so-called MAGIC biomarker profile that is based on only two markers, i.e., the proinflammatory acute phase biomarker IL33Rα/ST2 and the gastrointestinal marker Reg3α [90,91]. These studies showed that the initial response reflected in this algorithm after one week of steroid treatment correlated with the treatment response after 28 days as well the one-year nonrelapse mortality and one-year survival. Recently they also showed that the day 28 score in patients without signs of acute GVHD could be used to discriminate between patients with a low risk of later GVHD (9% with later acute GVHD) and a higher risk of later relapse (24% general risk but 33% risk in patients with high risk disease) [92]. For such patient it will then be reasonable to rapidly decrease the immunosuppressive GVHD prophylaxis and, thereby, increase antileukemic immune reactivity and decrease the risk of later relapse.

Taken together all these studies illustrate that although the acute phase reaction is a complex reaction that shows heterogeneity even between patients with the same diagnosis, it was possible to develop a very simple prognostic tool for patients with very complex inflammatory complications/disorders based on a careful selection of biomarkers and a standardization of the time of testing.

### 4.2. The Use of Inflammatory Markers to Identify Responders to Anticancer Immunotherapy and/or Patients with Increased Risk of Severe Immune-Mediated Toxicity

Various forms of anticancer immunotherapy are now tried in the treatment of human malignancies, including CAR-T cells, BITE antibodies, and checkpoint inhibitors [93]. Two main questions appear with regard to the acute phase reaction in patients receiving these new anticancer immunotherapies: (i) do pretreatment CRP and/or other conventional proinflammatory markers have a prognostic impact with regard to responsiveness/survival also in patients receiving this kind of treatment; and (ii) can pretreatment inflammatory markers be used to predict the toxicity of immunotherapy, and especially severe cytokine storm?

Several previous studies have investigated whether the development of immune-mediated toxicity after anticancer immunotherapy is associated with survival [94]. Five of these studies described associations between toxicity and improved overall survival [95,96] or response to immunotherapy [97,98,99] for patients with immune-mediated toxicity. This is similar to GVHD in allotransplant recipients; both immune-mediated adverse events for cancer patients and GVHD for allotransplant recipients are associated with anticancer immune effects but in severe forms both become life-threatening due to general toxic effects.

Increased pretherapy serum CRP levels are associated with adverse prognosis both with regard to response to treatment and survival of cancer patients receiving immunotherapy [100,101,102,103,104,105,106,107,108,109,110], but increased pretherapy CRP levels are also associated with an increased risk of immune-mediated adverse reactions, including severe cytokine release syndrome (Table 5) [103,109,110]. An increase in CRP levels may also precede the clinical symptoms of cytokine release syndrome [111].

The therapeutic interventions in patients with severe immune-related adverse events during anticancer immunotherapy include the use of anti-IL6 therapy [112]. This is an example on how the characterization of the acute phase cytokine response can be used as a scientific basis for recommended therapeutic interventions. IL6 targeting is also used in the treatment of GVHD; JA2 inhibition is used in the treatment of acute GVHD and this represents inhibition of JAK2-STAT3 signaling that is the main intracellular pathway downstream to the IL6Rα-gp130 receptor [113,114,115].

## 5. Discussion

In the present article, we have reviewed important aspects of the hematological experience with regard to the acute phase reaction in clinical oncology. This experience shows that the phenotype of the acute phase reaction can differ between patients, the reaction is observed and can be used as a biomarker even in severely immunocompromised cancer patients, it can be both a diagnostic and prognostic tool in clinical oncology, and further characterization of the reaction can be a basis for future studies of therapeutic interventions.

One of the remaining questions is whether the associations between CRP and prognosis reflects a role of CRP in carcinogenesis or whether it should be regarded as only a biomarker for inflammation. The conclusion from a recent review was that the available data suggest that circulating levels of CRP do not cause cancer, but epidemiological studies suggest that it is a marker of increased cancer risk and, as described in this review, it may then reflect important biological characteristics of human malignancies and thereby have a prognostic impact [116].

The acute phase reaction is a response to inflammation, injury, or infection. Intensified anticancer treatment is now increasingly used even in elderly patients; this is true even for the most aggressive diseases requiring very intensive treatment like allogeneic stem cell transplantation [74]. Intensified treatment will usually increase the risk of severe neutropenia and such severely immunocompromised patients are prone to a wide range of viral, bacterial, and fungal infections [117,118,119,120,121,122]. CRP levels can be used for the evaluation of anti-infectious treatment. However, a microbiological diagnosis is often not possible in these patients, is only possible by using invasive diagnostic procedures, or is not available at the time when anti-infectious treatment has to be started. In our opinion, future studies should investigate whether different infections are associated with different phenotypes (i.e., biomarker profiles) of the acute phase reaction, and whether this can be used for an early microbiological prediction in cancer patients. A recent study of immunocompetent patients showed that a limited number of acute phase mediators could be used as a biomarker of bacteremia in immunocompetent patients with sepsis [25]. In our opinion, this last observation suggests that acute phase reaction profiling should be further investigated as a possible biomarker of specific infections in cancer patients.

The peripheral blood monocyte levels have been investigated as a possible prognostic marker in relatively few studies (see Table 3) [43]. These studies used the total monocyte cell count. Several CRP isoforms can influence the functional characteristics of monocytes, including their capacity of proinflammatory cytokine release but also phagocytosis, differentiation, migration, and metabolism (Table 2). However, recent studies have demonstrated that circulating monocytes constitute a heterogeneous cell population; the main subsets are classical, intermediate, and non-classical monocytes but especially the intermediate subset seems to be heterogeneous [123,124,125,126,127,128]. The three main subsets can be identified by standardized flow-cytometric methods that are suitable for clinical implementation [123]. The majority of circulating monocytes in healthy individuals are the classical monocytes [123,128], but the levels of various subsets are altered in patients with inflammatory or malignant diseases [125,126,127,128]. Furthermore, total monocyte levels normalize early after both allogeneic and autologous stem cell transplantation, but patients differ with regard to the early reconstitution of the three main monocyte subsets [129,130]. Experimental studies have shown that even closely related pharmacological agents can differ in their effects on normal monocyte functions [131], and futures studies therefore have to clarify whether functional differences of anticancer agents between patients reflect differences in circulating monocyte subset levels. In our opinion, characterization of circulating monocyte subsets should be further investigated as a possible prognostic biomarker in cancer patients, including patients receiving allogeneic stem cell transplantation and immunotherapy. One of the reasons for this is the functional effects of various CRP isoforms on monocytes, and the question whether these CRP effects differ between monocyte subsets.

Inflammatory biomarkers have a prognostic impact on several malignancies; mainly solid tumors (see Table 3). However, the reviewed experience from multiple myeloma shows that evaluation of the acute phase reaction or other acute phase reactants may also become useful in hematological malignancies that often are referred to as liquid tumors. Future studies should in our opinion try to clarify whether acute phase biomarkers have a prognostic impact also on other hematological malignancies, especially various lymphoproliferative diseases. However, cancer cell-specific biomarkers, such as cytogenetic and molecular genetic abnormalities, will probably be most important for the high-risk hematological malignancies, such as acute leukemias and myelodysplastic syndromes, and normal peripheral blood cell counts will not be suitable markers in chronic myeloproliferative neoplasia because increased peripheral blood counts of leukocytes/erythrocytes/platelets then represent a disease characteristic and not a systemic reaction. Finally, if peripheral blood cell counts are used as biomarkers in hematological malignancies they have to be interpreted with caution because their levels may reflect both bone marrow involvement and a systemic inflammatory response [66].

Allogeneic stem cell transplantation is used in the treatment of aggressive hematological malignancies, and antileukemic effects can then be mediated both as a leukemia-specific graft versus leukemia reactivity and as a consequence of the general GVHD reaction directed towards antigens that are expressed both by normal and malignant recipient cells [132,133]. Thus, controlled GVHD reactivity is an advantage because of its antileukemic effect and the reduced risk of leukemia relapse, whereas an uncontrolled GVHD can be life threatening due to organ damage. The situation in anticancer immunotherapy is similar; a controlled general increase in immune reactivity mediates important anticancer effects whereas an uncontrolled general increase in the reactivity may lead to severe toxicity. In both cases, the pretreatment evaluation of the risk of toxicity versus the possibility of an anticancer immune effect is important. Thus, optimal early diagnosis and early intervention is important in both anticancer immune therapy and posttransplant acute GVHD. A similar strategy as used during the last decade for identification of clinically useful biomarkers in acute GVHD should probably be tried also in patient receiving anticancer immunotherapy.

Will CRP have a role in future cancer treatment? In our opinion, the answer is yes. CRP is already a useful biomarker especially for prognostication and diagnosis of complications to anticancer treatment. However, we believe that CRP may become even a more useful diagnostic or prognostic biomarker if it can be used in combination with other acute phase markers as a part of an acute phase profile. The biological functions of CRP are now being explored, and some studies suggest that CRP may even become a therapeutic target for anti-inflammatory treatment [35].

Recent reviews of epidemiological studies suggest that inflammation can predispose to cancer, and targeting of inflammation and the molecular mechanisms involved in the inflammatory processes may therefore represent a possible strategy for cancer prevention [134,135]. This is also supported by previous studies suggesting that nonsteroidal anti-inflammatory drugs (NSAID) and the more specific COX2 inhibitors can be used for cancer prevention [136,137]. The mechanisms behind these effects of inflammation and inhibition of inflammation can be direct effects on the malignant cells, but indirect effects are also possible because the anti-inflammatory agents can interfere with the cancer cell microenvironment, including various immunocompetent cells [134,135,137]. Such chemopreventive strategies seem to be relevant in a wide range of malignancies [134]. However, previous studies show that the inflammatory phenotype (i.e., the acute phase protein/cytokine profile) differs between patients with the same malignant disease and between different malignancies [72,73]. Several anti-inflammatory agents seem to have such a chemopreventive effect. In our opinion, future studies should therefore investigate whether the optimal chemopreventive agent differs between patients and/or between anti-inflammatory agents.

## 6. Conclusions

What can we then learn from the hematological experience with regard to the acute phase reaction in clinical oncology? First, broad acute phase (cytokine) responses can be detected even in severely immunocompromised patients receiving the most intensive anticancer therapy. Second, hematological biomarkers of inflammation (i.e., peripheral blood cell counts of leukocytes/erythrocytes/platelets) can be used together with molecular acute phase biomarkers for prognostication and therapy monitoring in cancer patients. Alternative markers should be further investigated as markers of the acute phase reaction, including systemic levels of alternative single molecules, biomarker profiles, and monocyte subsets. The optimal biomarker or biomarker combination to use for prognostication or therapy monitoring will probably differ between various malignancies and possibly also between patient subset with the same malignant disease. Third, the GVHD experience shows that based on an initial screening of several inflammatory biomarker it is possible to identify a few optimal biomarkers and an optimal time point for prognostic evaluation of patients. A similar strategy should be tried to identify relevant biomarkers to use in patients receiving anticancer immunotherapy. A major goal would then be to try to identify proinflammatory biomarkers that can distinguish between the likelihood of an anticancer effect versus the risk of severe immune-mediated complications. The possibility of combining inflammatory biomarkers and organ-specific markers should also be considered in anticancer immunotherapy.

## Figures and Tables

**Table 1 cancers-12-01966-t001:** The acute phase reaction in leukemia patients with febrile neutropenia; a summary of important studies in patients with hematological malignancies and severe chemotherapy-induced pancytopenia in peripheral blood due to chemotherapy-induced bone marrow failure. The patients included in most of these studies had fever (>38.5 °C), increasing C-reactive protein (CRP) levels to >50 mg/L and either documented or likely bacterial infections [10,11,12,13,14,15,16,17,18,19,20,21,22,23,24] ^1^.

Classification	Mediators	Altered Level During Infection
Cytokines	Chemokines	Increased levels of CCL2, CXCL8/IL8 Decreased levels of CCL5
	Interleukins	Increased levels of IL1β, IL4, IL5, IL6, IL8, IL10
	Growth factors	Increased levels of G-CSF, GM-CSF, thrombopoietin
	Immunomodulators	Increased levels of TNFα and IFNγ
	Cytokine receptors or antagonists	Increased levels of soluble IL4Rα Increased levels of Type I and type II TNF receptors Increased levels of IL1RA
Soluble cell surface molecules	Soluble triggering receptor expressed on myeloid cells	Increased
	Selectins	Decreased levels of soluble E-selectin (expressed by endothelial cells), P-selectin (platelets, megakaryocytes, endothelium) and L-selectin
	ICAM1, CD14	Increased levels
Proteases	MMP10, TIMP1	Increased levels
Matrix molecules		Increased levels of endocan (endothelium-derived)
Other markers		Increased levels of phospholipase-A2, whereas levels of elastase (neutrophil marker) and neopterin (marker of monocyte activation) are not altered Decreased levels of albumin and Fas-ligand

^1^ Abbreviations: G-CSF, granulocyte colony stimulating factor; GM-CSF, granulocyte-macrophage colony-stimulating factor; ICAM, intercellular adhesion molecule; IFN, Interferon; IL, interleukin; IL1RA, IL1 receptor antagonist; MMP, matrix metalloprotease; TIMP, tissue inhibitor of metalloprotease; TNF, tumor necrosis factor.

**Table 2 cancers-12-01966-t002:** A summary of important molecular and biological characteristics of CRP. For more detailed information and additional references, we refer to several recent review articles [7,9,26,27]; additional original articles describing specific observations are given in the table ^1^.

Characteristic	Description
Baseline levels	Influenced by several factors including age, gender, smoking, weight, blood pressure, lipid levels, CRP gene polymorphisms, hormone replacement therapy
Isoforms	Native CRP is a pentameric protein. Monomeric CRP: formed by irreversible dissociation of the pentamere into monomers (206 amino acids, molecular weight 23 kDa). CRP is synthesized as monomers; the pentamere is then formed in the endoplasmatic reticulum where it is stored and from where it is released slowly at the non-inflammatory state. The pentamere is rapidly released in response to increased levels of proinflammatory cytokines. CRP can also form fibril-like structures, decameres and possibly trimers and tetramers Various CRP peptides may also mediate biological effects [28].
CRP releasing cells	The native isoform is mainly released by hepatocytes but can also be released by smooth muscle cells, macrophages, endothelial cells, lymphocytes, and adipocytes.
CRP release	The pentamere is formed and stored in the endoplasmatic reticulum from where it is slowly released in the absence of inflammation. CRP is rapidly released in response to proinflammatory cytokines. When the inflammatory stimulation is removed CRP levels decrease with a half-life of 18–20 h.
CRP gene expression	IL6 is important for CRP expression but is not sufficient alone; TNFα, IL8/CXCL8, and CCL2 (MCP1) can also stimulate CRP expression.
CRP ligands	The CRP pentamere can bind to a wide range of ligands including polysaccharides, proteins, chromatin/nuclear antigens, damaged cell membranes, apoptotic cells.
CRP receptors	The complement receptors FcγRI (CD64), FcγRIIa (CD32), FcαRI (CD89), Lectin-like oxidized low-density lipoprotein receptor-1 (LOX-1), αvβ3, and α4β1 integrins, FcγRIII (CD16), lipid rafts.
Important pentameric CRP effects	Complement: activation of the classical complement cascade Monocytes/macrophages: polarization to the proinflammatory M1 phenotype, stimulation of phagocytosis and cytokine release, inhibition of chemotaxis, increased LDL uptake. Dendritic cells: CRP seems to be an important regulator even in the absence of an acute phase response; it can also activate monocyte-derived dendritic cells and thereby induce T cell activation [29,30]. Neutrophils: inhibition of activation and chemotaxis, stimulation of phagocytosis depending on the biological context. Endothelial cells: activation. Platelets: inhibition of activation, trafficking, and aggregation.
Important monomeric CRP effects	Monocytes: stimulated reactive oxygen species (ROS) release. Neutrophils: induced activation/adherence/ trafficking; reduced apoptosis Endothelial cells: activation.
CRP peptide AA 201-206	Inhibition of neutrophil adhesion to endothelial cells; inhibition of platelet activation and capture of neutrophils [27].

^1^ Abbreviations: AA, amino acids; LDL, low-density lipoprotein.

**Table 3 cancers-12-01966-t003:** Markers of proinflammatory activity investigated as possible prognostic parameters in cancer patients [42,43,44,45]. The table summarizes observations from meta-analyses of the various proinflammatory markers investigated in various malignancies (mainly solid tumors).

Inflammatory Parameter	Comments/Observations
CRP	This is the most frequently studied proinflammatory parameter, and it has been investigated in several retrospective and prospective studies [43]. One of the available systematic reviews described the results for 271 articles [42]. Increased CRP was associated with an adverse prognosis and increased mortality in 245 of these studies, and for 80% of these studies the increased mortality was predicted in multivariate analyses. Half of the articles investigated patients with gastrointestinal or kidney malignancies.
Albumin	A systematic review identified 31 studies including 9753 patients [43]. The frequency of patients with albumin <30 g/L varied between 20 and 50% in different studies. Meta-analyses showed significant associations between low albumin and adverse survival both when using albumin cutoff of 30 and 35 g/L.
CRP:albumin ratio (Glasgow prognostic score)	Both the original Glasgow prognostic score and the modified Glasgow prognostic score were based on a cut-off value of <35 g/L for albumin and >10 mg/L for CRP [45]. An early review described associations between a high score and adverse prognosis both for patients with operable (28 studies, 8333 patients) and inoperable tumors (11 studies, 2119 patients); and similar associations may also be present for patients receiving chemoradiotherapy especially for colorectal and gastroesophageal cancer [45]. An updated meta-analysis of studies including patients without metastases (25 studies, 12,097 patents) showed an association between high pretreatment ratio and survival; colorectal cancer was an exception in this study [44].
White blood cell count	Relatively few studies have investigated this parameter, and to the best of our knowledge, no meta-analyses are available. Increased levels are seen for 20–30% of patients [43].
Neutrophil count	A recent systematic review and meta-analysis was based on nine publications including 2870 patients [43]. There was a significant association between granulocytosis and decreased survival. The frequency of patients with granulocytosis varied between studies (12–32%).
Lymphocyte count	A recent review was based on 11 articles (2517 patients) [43]. The overall data showed a significant association between low lymphocyte counts and adverse prognosis. However, there was a considerable variation in lymphocyte threshold between studies.
Monocyte count	A recent review identified five retrospective studies based on multivariate analyses including 1152 patients [43]. Monocytosis was associated with an adverse prognosis. The proportion of patients with monocytosis in these studies was above 20% and was 57% for the study with the highest proportion.
Platelet count	A recent review identified seven studies including 2293 patients, and they observed a general association between increased platelet counts and survival [43]. The proportion of patients with thrombocythemia was between 10% and 30% in most of these studies.
Neutrophil:lymphocyte ratio	A recent review was based on 59 articles (16,921 patients) [43]. Significant associations between increased levels and adverse prognosis (i.e., reduced overall survival) were observed both when using a threshold of 4 or 5 mg/L. The proportion of patients with CRP levels >5 mg/L was 20-50% for most studies.
Lymphocyte:monocyte ratio	A recent review described the results for 11 publications (5043 patients) and observed a significant association between a low ratio and adverse prognosis [43]. However, different thresholds were used, but usually approximately 50% of the patients had a low ratio.
Platelet:lymphocyte ratio	An increased ratio is relatively common in cancer patients (20–60%); single studies have described associations between this ratio and adverse prognosis but the threshold used varies considerably between studies and no meta-analyses are available [43].

**Table 4 cancers-12-01966-t004:** The prognostic impact of inflammatory mediators in cancer; a summary of the results from systematic reviews and meta-analyses [46,47,48,49,50,51,52,53,54,55,56,57,58,59,60,61,62,63,64] ^1^.

Malignancy	Number of Patients/Studies	Observation
Breast [46]	4502/10	CRP: significant association between CRP and overall, disease-free and cancer-specific survival
Osteosarcoma [47]	397/2	CRP: increased levels associated with adverse prognosis with reduced overall survival
Nasopharyngeal [48]	5215/5	CRP: increased levels associated with adverse prognosis
Lung cancer [49,50]	3165/10	CRP: high pretreatment levels were associated with poor overall survival
	1257/4	CRP:albumin ratio: levels associated with poor overall survival in multivariate analysis. The cut-off values varied between the four studies
Pancreatic [51,52,53]	685/10	Resectable pancreatic cancer: only a trend for adverse prognosis in some of the studies
	1804/11	Neutrophil:lymphocyte ratio: unfavorable impact on both overall and cancer-specific survival
	3182/15	Neutrophil:lymphocyte ratio: patients with a low ratio had better overall and progression-free survival
Esophageal [55]	4551/10	Neutrophil:lymphocyte and CRP:albumin ratios: high ratios were associated with poor overall survival
Colorectal [56,57]	3431/9	CRP:albumin ratio: high ratio associated with decreased overall survival; high values also correlated with large tumor diameter and lymph node metastases
	1705/12	CRP: increased levels associated with shorter overall and disease-free survival both for local and advanced disease
Urological [58]	7490/43	CRP: high CRP associated with reduced overall, cancer-specific, and relapse-free survival in for urological cancer. Furthermore, a review article suggests that this prognostic impact is seen both for renal cell, upper urinary, bladder, and prostatic cancer [65]
Bladder [59,60]	5546/34	CRP: high levels being an independent prognostic marker in urothelial bladder carcinoma
	22,224/32	Bladder cancer treated with radical cystectomy: decreased survival associated with high neutrophil:lymphocyte ratio, CRP and white blood cell count
Renal [61,62]	14,136/47	CRP, platelet count: increased levels associated with reduced overall and cancer-specific survival
	4100/24	CRP: increased levels associated with higher stage, grade, overall mortality, cancer-specific mortality, and cancer progression. Decreased survival was also seen for patients with local disease at the time of diagnosis
Prostate [63,64]	1497/5	CRP: high levels associated with overall and progression-free survival ^1^
	659/6	Metastatic prostate cancer: high CRP levels associated with decreased overall survival

**Table 5 cancers-12-01966-t005:** The use of proinflammatory markers for prognostication in patients receiving anticancer immunotherapy; a summary of the results from important large clinical studies [99,100,101,102,103,104,105,106,107,108,109] ^1^.

Prognostic Use and Therapeutic Agent	Comments
Response to treatment	
Ipilimumab [100]	Melanoma patients (*n* = 720): both pretreatment high absolute neutrophil count and high neutrophil:lymphocyte ratio were associated with adverse impact both with regard to disease progression and death, and with each of the two parameters the prognosis worsened (i.e., they had independent impact].
Ipilimumab [101]	Melanoma patients (*n* = 58): high pretreatment neutrophil:lymphocyte ratio was associated with reduced survival in multivariate analyses.
Ipilimumab [102]	Melanoma patients (*n* = 95): both disease control and survival was associated with decreasing CRP levels as well as decreasing levels of circulating regulatory T cells and increasing absolute lymphocyte counts during treatment, i.e., 12 weeks after initiation of treatment.
Ipilimumab [103]	Melanoma patients (*n* = 113): high pretreatment level of soluble CTLA4 had a favorable prognostic impact, was higher in responders to therapy, and was also associated with survival; these levels also increased during treatment. However, high levels were also associated with immune-related adverse events, especially of the gastrointestinal tract.
Atezolizumab [104]	Advanced pulmonary cancer (*n* = 751 + 797): pretreatment CRP level was associated with an adverse prognosis and the most predictive biomarker for overall and progression-free survival. CRP was included in a prognostic index together with LDH, PD-L1 expression, performance status, time since metastases and metastatic site count.
Nivolumab [105]	Metastatic renal cell carcinoma (*n* = 58): increased neutrophil:lymphocyte, monocye:lymphocyte and platelet:lymphocyte ratios were associated with shorter progression-free survival, and overall survival was significantly shorter for patients with increased levels of these three ratios as well as for patients with high CRP. The monocyte:lymphocyte ratio was an independent factor for progression-free survival, whereas high monocyte ratio, neutrophil:lymphocyte ratio and CRP levels were independent factors for overall survival.
Nivolumab or pembrolizumab [106]	Non-small cell lung cancer (*n* = 34): samples were collected within 7 days before and 7 days after initiation of therapy. After initiation of treatment the IL6 and CRP levels increased for a subset of patients, and this subset showed an increased frequency of response to treatment and a prolonged overall survival.
Avelumab, nivolumab, pembrolizumab [107]	Gastric/gastroesophageal cancer (*n* = 57): several proinflammatory markers were tested. High neutrophil:lymphocyte ratio, high CRP, and low albumin were all associated with short overall survival and were built into a prognostic nomogram. Patients with high score had a survival of only few months, patients with low score lived for more than 16 months.
Risk of toxicity	
Ipilimumab [103]	Melanoma patients (*n* = 113): high pretreatment levels of soluble CTLA4 had a favorable prognostic impact (see above) but were also associated with the risk of immune-related events, especially of the gastrointestinal tract.
CAR-modified T cells with anti-CD19 [108]	Relapsed/refractory ALL (*n* = 51): the peak cytokine levels were determined during the first month after infusion, and peak levels of IL6, IL8, sIL2Rα, sIL6R, IFNγ, CCL2, CCL3, CCL4, sgp130, and GM-CSF. Later development of severe cytokine release syndrome could be predicted both in the test and validation group based on a signature including three cytokines. CRP levels during the same period were not predictive but increased during severe CRS. Another study has confirmed that CRP can serve as an indicator of severity for CRS [110].
Ipilimumab [109]	Melanoma (*n* = 140): a severe adverse event was observed for 36 patients. The authors investigated cytokine levels, white blood cell counts, and tumor burden parameters. Females showed an increased frequency of adverse event, but associations were observed between low pretherapy IL6 levels and higher overall survival as well as higher risk of adverse events.

^1^ Abbreviations: ALL, acute lymphoblastic leukemia; CRS, cytokine release syndrome.

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
