# Peer review of "Combined C-Reactive Protein and Novel Inflammatory Parameters as a Predictor in Cancer—What Can We Learn from the Hematological Experience?"

_cancers, 2020, doi:10.3390/cancers12071966_

Round 1

Reviewer 1 Report

Øystein Bruserud et al. comprehensively reviewed the clinical predictive values of C-reactive Protein (CRP) and various acute phase reactants (APR) for myriads of hematological malignancies. This study topic is interesting and important, attributing to Prof. Bruserud’s long-term efforts and contributions in this scientific field. Although the article is well-written, I have a number of comments concerning this study:

  1. In light of conclusion of the study “future studies should not only investigate the conventional proinflammatory mediators (e.g. CRP, white blood cell counts) but also levels of alternative regulators of inflammation, biomarker profiles, and combinations of inflammatory mediators with specific organ markers”, the title of the article could be rephrased as “Combined CRP and novel inflammatory parameters as a predictor in cancer–What Can We Learn from the HEMATOLOGICAL experience?” to highlight the scope of the study.
  2. Theoretically, there should be no significant change in the circulating concentration of APR if chronic inflammation is in the steady state. Line 11: “a systemic response to acute or chronic inflammation” might be rephrased as “a systemic response to acute inflammation or chronic state with acute reaction”.
  3. Line 12: The word “molecular” and Line 13 “CRP is released mainly by hepatocytes but can by various other cells/organs” seem redundant in Abstract and should be deleted.
  4. Line 13: It could be rephrases as “…, including its uses for prognostics and therapy-monitoring in cancer patients.”
  5. Line 14: The words “also”, “other”, “biomarker”, “cytokines” and “but” occur too often throughout the manuscript. It could be rephrased as “Although Interleukin 6 (IL 6) is a main trigger of the acute phase reactions, a series of acute phase reactants (APR) can contribute (e.g. other members in IL 6 family, IL1 subfamily, and tumor necrosis factor α). The statement “the main trigger” is too strong.
  6. Line 18 Rephrasing the sentence: …shown that, besides CRP, other biomarkers (e.g. cytokines, soluble cytokine mediators, soluble adhesion molecules)…
  7. Line 20 Rephrasing the sentence: Furthermore, CRP and white blood cell counts serve as a dual predictor in solid tumors and hematological malignancies. Recent studies suggest that biomarker profiles as well as alternative inflammatory mediators should be further developed to optimize the predictive utility in cancer patients.
  8. Line 25 Rephrasing the sentence: selected acute phase reactants (APR) together with specific markers of organ damages are useful for predicting or diagnosing graft versus host disease. APR may also be useful to identify patients (at risk of) developing severe immune-mediated toxicity after anticancer immunotherapy.
  9. Line 28 Rephrasing the sentence: future studies of molecular predictors in human malignancies should not only investigate the conventional inflammatory mediators (e.g. CRP, white blood cell counts) but also combinations of novel inflammatory parameters with specific markers of organ damages.
  10. Line 37 Rephrasing the sentence: a marker of the acute phase reaction (APR) that can be induced both by acute and chronic states with a systemic reaction to of inflammation, injury or infections [1-2].
  11. Line 37 Rephrasing the sentence: APR alters circulating levels of a myriad of inflammatory mediators [3-5], including certain cytokines that can functions as inducers or drivers of the reaction and/or can be released during the later phase [1].
  12. Line 41 Rephrasing the sentence: CRP is the only acute phase biomarker that is used in routine clinical practice [3, 6, 7], whilst several circulating biomarkers of inflammatory have been characterized during the last decade.
  13. Line 43 Rephrasing the sentence: These new parameters may serve as an alternative to CRP or should be combined
  14. Line 49 Rephrasing the sentence: …of several acute phase proteins in response to inflammation, infection or tissue injury, but the name is misleading because the reaction contains chronic diseases with fluctuating conditions, e., cancers.
  15. Line 60-61: The sentence should be rephrased.
  16. Line 65 Rephrasing the sentence: several proteins, e.g. albumin, transferrin and IGF1, can be a part of the acute phase reaction and associated with altered peripheral blood cell counts
  17. Line 71 rephrase: …and altered serum levels of associated proteins responding to inducers.
  18. Line 82-83: The sentence should be rephrased.
  19. Line 93: The word “only” should be deleted.
  20. Line 128: immunoregulatory mediator => immunoregulator or immunomodulatory
  21. Line 185-186: “An alternative to CRP/albumin is to use normal peripheral blood cell counts, either as the cell count alone or as a cell count ratio.” The sentence makes me confused.
  22. Line 240: “The peripheral blood normal cell counts are more difficult to use for prognostication in patients with hematological malignancies because these diseases will often have bone marrow involvement that will influence the interpretation of these cell counts.” Does dehydration (or steroid use, etc.) have a great influence on peripheral blood normal cell counts?
  23. Line 374: CRP serum levels => serum CRP levels
  24. Line 401: Intensified anticancer treatment is now increasingly used, even in elderly patients. This is true…
  25. Line 437 and 441: impact “on”
  26. Line 437: acute phase reaction or other proinflammatory markers => acute phase reactants that often are referred to as liquid tumors may also become useful in hematological malignancies.
  27. Line 446: The sentence “increased normal cell counts then represent a disease characteristic and not a systemic reaction.” is confusing.
  28. Line 448: with “caution”
  29. Line 471: a broad acute phase (cytokine) response=> responses
  30. Line 473: hematological biomarkers of inflammation (i.e. normal peripheral blood cell counts)…In my opinion, “normal” peripheral blood can’t stand for biomarkers of inflammation.
  31. The conclusions section is very important in the article that should be refined.
  32. prognostication of cancer => prognostication and therapy-monitoring in cancer

Author Response

ALL ALTERATIONS IN THE REVISED VERSION OF THE MANUSCRIPT ARE MARKED WITH YELLOW

REVIWER 1

SUMMARY: All suggestions have been incorporated in the Revised Version except for comment 2; please see our response below.

Øystein Bruserud et al. comprehensively reviewed the clinical predictive values of C-reactive Protein (CRP) and various acute phase reactants (APR) for myriads of hematological malignancies. This study topic is interesting and important, attributing to Prof. Bruserud’s long-term efforts and contributions in this scientific field. Although the article is well-written, I have a number of comments concerning this study:

  1. In light of conclusion of the study “future studies should not only investigate the conventional proinflammatory mediators (e.g. CRP, white blood cell counts) but also levels of alternative regulators of inflammation, biomarker profiles, and combinations of inflammatory mediators with specific organ markers”, the title of the article could be rephrased as “Combined CRP and novel inflammatory parameters as a predictor in cancer–What Can We Learn from the HEMATOLOGICAL experience?” to highlight the scope of the study.

The title has been changed.

  1. Theoretically, there should be no significant change in the circulating concentration of APR if chronic inflammation is in the steady state. Line 11: “a systemic response to acute or chronic inflammation” might be rephrased as “a systemic response to acute inflammation or chronic state with acute reaction”.

In our opinion one may also include the possibility where you have increased levels above the normal range over time as a long-term reaction in response to a chronic disease. However, due to the overall length of the abstract we have not included such a detailed explanation. This was a suggestion made by the reviewer, and we hope it is acceptable to keep the original short statement.

  1. Line 12: The word “molecular” and Line 13 “CRP is released mainly by hepatocytes but can by various other cells/organs” seem redundant in Abstract and should be deleted.

We have made theses corrections.

  1. Line 13: It could be rephrases as “…, including its uses for prognostics and therapy-monitoring in cancer patients.”

The correction has been made.

  1. Line 14: The words “also”, “other”, “biomarker”, “cytokines” and “but” occur too often throughout the manuscript. It could be rephrased as “Although Interleukin 6 (IL 6) is a main trigger of the acute phase reactions, a series of acute phase reactants (APR) can contribute (e.g. other members in IL 6 family, IL1 subfamily, and tumor necrosis factor α). The statement “the main trigger” is too strong.

The correction has been made.

  1. Line 18 Rephrasing the sentence: …shown that, besides CRP, other biomarkers (e.g. cytokines, soluble cytokine mediators, soluble adhesion molecules)…

This has been done.

  1. Line 20 Rephrasing the sentence: Furthermore, CRP and white blood cell counts serve as a dual predictor in solid tumors and hematological malignancies. Recent studies suggest that biomarker profiles as well as alternative inflammatory mediators should be further developed to optimize the predictive utility in cancer patients.

This has been done.

  1. Line 25 Rephrasing the sentence: selected acute phase reactants (APR) together with specific markers of organ damages are useful for predicting or diagnosing graft versus host disease. APR may also be useful to identify patients (at risk of) developing severe immune-mediated toxicity after anticancer immunotherapy.

This has been done.

  1. Line 28 Rephrasing the sentence: future studies of molecular predictors in human malignancies should not only investigate the conventional inflammatory mediators (e.g. CRP, white blood cell counts) but also combinations of novel inflammatory parameters with specific markers of organ damages.

This has been done.

  1. Line 37 Rephrasing the sentence: a marker of the acute phase reaction (APR) that can be induced both by acute and chronic states with a systemic reaction to of inflammation, injury or infections [1-2].

This has been done.

  1. Line 37 Rephrasing the sentence: APR alters circulating levels of a myriad of inflammatory mediators [3-5], including certain cytokines that can functions as inducers or drivers of the reaction and/or can be released during the later phase [1].

This has been done.

  1. Line 41 Rephrasing the sentence: CRP is the only acute phase biomarker that is used in routine clinical practice [3, 6, 7], whilst several circulating biomarkers of inflammatory have been characterized during the last decade.

This has been done.

  1. Line 43 Rephrasing the sentence: These new parameters may serve as an alternative to CRP or should be combined.

The sentence has been rewritten.

  1. Line 49 Rephrasing the sentence: …of several acute phase proteins in response to inflammation, infection or tissue injury, but the name is misleading because the reaction contains chronic diseases with fluctuating conditions, e., cancers.

This has been done.

  1. Line 60-61: The sentence should be rephrased.

This has been done.

  1. Line 65 Rephrasing the sentence: several proteins, e.g. albumin, transferrin and IGF1, can be a part of the acute phase reaction and associated with altered peripheral blood cell counts

This sentence has been rewritten.

  1. Line 71 rephrase: …and altered serum levels of associated proteins responding to inducers.

The sentence has been rewritten.

  1. Line 82-83: The sentence should be rephrased.

The last part of this chapter has been rewritten.

  1. Line 93: The word “only” should be deleted.

This is corrected as suggested by the reviewer.

  1. Line 128: immunoregulatory mediator => immunoregulator or immunomodulatory

This is corrected as suggested by the reviewer.

  1. Line 185-186: “An alternative to CRP/albumin is to use normal peripheral blood cell counts, either as the cell count alone or as a cell count ratio.” The sentence makes me confused.

This sentence has been rewritten.

  1. Line 240: “The peripheral blood normal cell counts are more difficult to use for prognostication in patients with hematological malignancies because these diseases will often have bone marrow involvement that will influence the interpretation of these cell counts.” Does dehydration (or steroid use, etc.) have a great influence on peripheral blood normal cell counts?

These patients have no or only weak symptoms and may have their disease for several years before it is diagnosed. A brief additional comment is now included in the text to explain this.

  1. Line 374: CRP serum levels => serum CRP levels
  1. This is corrected as suggested by the reviewer.
  1. Line 401: Intensified anticancer treatment is now increasingly used, even in elderly patients. This is true…
  1. This is corrected as suggested by the reviewer.
  1. Line 437 and 441: impact “on”

This is corrected as suggested by the reviewer.

  1. Line 437: acute phase reaction or other proinflammatory markers => acute phase reactants that often are referred to as liquid tumors may also become useful in hematological malignancies.

This is corrected as suggested by the reviewer.

  1. Line 446: The sentence “increased normal cell counts then represent a disease characteristic and not a systemic reaction.” is confusing.

This is now rewritten we refer to the normal cells, i.e. leukocytes/erythrocytes/platelets.

  1. Line 448: with “caution”

This is corrected as suggested by the reviewer.

  1. Line 471: a broad acute phase (cytokine) response=> responses

This is corrected as suggested by the reviewer.

  1. Line 473: hematological biomarkers of inflammation (i.e. normal peripheral blood cell counts)…In my opinion, “normal” peripheral blood can’t stand for biomarkers of inflammation.

This is now rewritten we refer to the normal cells, i.e. leukocytes/erythrocytes/platelets.

  1. The conclusions section is very important in the article that should be refined.
  2. prognostication of cancer => prognostication and therapy-monitoring in cancer

This is corrected as suggested by the reviewer.

Reviewer 2 Report

The present review article describes the C reactive protein (CRP),  the only molecular acute phase biomarker used in routine clinical practice, including in cancer patients. Biomarkers of inflammation, including CRP can also be prognostic markers in solid tumors as well as hematological malignancies, and recent studies suggest that biomarker profiles as well as several alternative single mediators should be further investigated as possible prognostic biomarkers in cancer patients. The experience from allogeneic stem cell transplantation suggests that the combination of selected acute phase biomarker together with specific markers for organ damage might be useful for predicting or diagnosing graft versus host disease. The acute phase reaction may also be useful to identify patients (at risk of) developing severe immune-mediated toxicity after anticancer immunotherapy.

The review is well written, clear and deep enough to give to the reader a clear scenario on the role and the relevance of CRP. The only minor suggestion is to further improve the conclusions which might better indicate the critical opinion of the authors and the perspectives.

Author Response

ALL ALTERATIONS IN THE REVISED VERSION OF THE MANUSCRIPT ARE MARKED WITH YELLOW

REVIEWER 2

The present review article describes the C reactive protein (CRP),  the only molecular acute phase biomarker used in routine clinical practice, including in cancer patients. Biomarkers of inflammation, including CRP can also be prognostic markers in solid tumors as well as hematological malignancies, and recent studies suggest that biomarker profiles as well as several alternative single mediators should be further investigated as possible prognostic biomarkers in cancer patients. The experience from allogeneic stem cell transplantation suggests that the combination of selected acute phase biomarker together with specific markers for organ damage might be useful for predicting or diagnosing graft versus host disease. The acute phase reaction may also be useful to identify patients (at risk of) developing severe immune-mediated toxicity after anticancer immunotherapy.

The review is well written, clear and deep enough to give to the reader a clear scenario on the role and the relevance of CRP. The only minor suggestion is to further improve the conclusions which might better indicate the critical opinion of the authors and the perspectives.

Response:

We are very grateful for this general comment. We have carefully revised our conclusion.

Reviewer 3 Report

Thank you very muh for the possibility to review the manuscript titled "C-reactive Protein as a Biomarker in Cancer- what can we learn from the hematological Experience?" written by Bruserud and colleagues.

In my opinion this is a very well written review dealing with the clinical relationship between C-reactive protein (CRP) and the different types of cancer. 

However I just have two minor queries:

  • Line 222: I think this would be better ....." The overall results also show that ELEVATED proinflammatory markers are generally associated with adverse prognosis and decreased survival"....
  • To round off this extensive topic the authors should discuss the relationship between CRP and the effect of anti-inflammatory agents (i.e. NSAR) with regard to the development and prevention of cancer. In my opinion this short outlook would be very interesting and therefore the paper would gain in importance.

Author Response

ALL ALTERATIONS IN THE REVISED VERSION OF THE MANUSCRIPT ARE MARKED WITH YELLOW

REVIEWER 3

  • Line 222: I think this would be better ....." The overall results also show that ELEVATED proinflammatory markers are generally associated with adverse prognosis and decreased survival"....

Response:

This has now been corrected as suggested by the reviewer.

  • To round off this extensive topic the authors should discuss the relationship between CRP and the effect of anti-inflammatory agents (i.e. NSAR) with regard to the development and prevention of cancer. In my opinion this short outlook would be very interesting and therefore the paper would gain in importance.

Response:

A new chapter with new references has been added at the end of the Discussion section.